# Effects of Hyperbaric Oxygen Therapy on Hemogram, Serum Biochemistry and Coagulation Parameters of Dogs Undergoing Elective Laparoscopic-Assisted Ovariohysterectomy

**DOI:** 10.3390/ani14121785

**Published:** 2024-06-14

**Authors:** Bernardo Nascimento Antunes, Pâmela Caye, Otávio Henrique de Melo Schiefler, Jenifer Jung, João Segura Engelsdorff, Vitória Pina Tostes, Emanuelle Bortolotto Degregori, Rainer da Silva Reinstein, Cinthia Melazzo De Andrade, Maurício Veloso Brun

**Affiliations:** 1Graduate Program in Veterinary Medicine, Center of Rural Science, Federal University of Santa Maria (UFSM), Av. Roraima, 1000, Building 42, Room 3135, Santa Maria 97105-900, RS, Brazil; pamiscaye@gmail.com (P.C.); vetotavio@gmail.com (O.H.d.M.S.); mvjeniferjung@gmail.com (J.J.); joaoengelsdorff@hotmail.com (J.S.E.); emanuelle.bortolotto@gmail.com (E.B.D.); rainerreinstein@gmail.com (R.d.S.R.); cmelazzoandrade1@gmail.com (C.M.D.A.); mauriciovelosobrun@hotmail.com (M.V.B.); 2Department of Small Animal Clinics, Center of Rural Science, Federal University of Santa Maria (UFSM), Av. Roraima, 1000, Building 42, Room 3135, Santa Maria 97105-900, RS, Brazil; pinatostesvitoria@gmail.com

**Keywords:** canine, hemostasis, monoplace hyperbaric chamber, oxygen therapy

## Abstract

**Simple Summary:**

Hyperbaric oxygen therapy (HBOT) has broad potential as an adjuvant therapy for various medical and surgical conditions with potential preoperative use in the preservation and/or preparation of the surgical bed. However, further studies on their effects on healthy individuals are required. This study explored the effects of HBOT on hemogram, serum biochemistry and hemostatic parameters in female dogs undergoing laparoscopic-assisted ovariohysterectomy. Thirty adult, mixed-breed, healthy female dogs were randomly separated into three groups: HBOT + SURG group (exposure to two ATAs for 45 min followed by video-assisted OVH), HBOT group (exposure to two ATAs for 45 min) and SURG group (video-assisted OVH). Blood samples were collected at T0 (at the admission), at T1, 24 h after T0 (immediately after HBOT in the HBOT + SURG and HBOT groups, and immediately before anesthetic premedication in the SURG group), and at T2, 48 h after T0 (24 h after HBOT and anesthetic premedication). Laboratory assessments included erythrocyte, leukocyte and platelet count, renal and hepatic serum biochemistry, prothrombin time (PT), activated partial thromboplastin time (APTT), buccal mucosal bleeding time (BMBT) and bloodstain area in a hygroscopic paper (BA). In conclusion, a session of HBOT at two ATAs for 45 min did not cause changes in the BMBT and BA of healthy bitches. Some modifications in leukocyte, neutrophil and lymphocyte count, as well as in alkaline phosphatase, PT and APTT were observed in the different groups considered, mostly dependent on the use of HBOT.

**Abstract:**

Background: This study explored the effects of hyperbaric oxygen therapy (HBOT) on hemogram, serum biochemistry and hemostatic variables in female dogs undergoing laparoscopic-assisted ovariohysterectomy (OVH). Materials: Thirty adult, mixed-breed, healthy female dogs were randomly divided into the following three groups: HBOT + SURG (exposed to two absolute atmospheres (ATAs) for 45 min followed by laparoscopic-assisted OVH), HBOT (exposed to two ATAs for 45 min) and SURG (laparoscopic-assisted OVH). Blood samples were collected at T0 (at the admission), at T1, 24 h after T0 (immediately after HBOT in the HBOT + SURG and HBOT groups, and immediately before anesthetic premedication in the SURG group), and at T2, 48 h after T0 (24 h after HBOT and anesthetic premedication). Methods: Assessments included erythrogram, leukogram, thrombogram, renal and hepatic serum biochemistry, prothrombin time (PT), activated partial thromboplastin time (APTT), buccal mucosal bleeding time (BMBT) and bloodstain area (BA) on hygroscopic paper collected at the BMBT. Results: Both the HBOT + SURG and SURG groups presented neutrophilia (*p* ≤ 0.0039) at T2 and an increase of ALP at T2 (*p* ≤ 0.0493), the SURG group presented an increase in leukocyte count at T2 (*p* = 0.0238) and the HBOT + SURG group presented a reduction in lymphocyte count at T2 (*p* = 0.0115). In the HBOT + SURG group, there was a reduction in PT and APTT in relation to the baseline value (*p* ≤ 0.0412). Conclusions: A session of HBOT at two ATAs for 45 min did not cause changes in the BMBT or BA in healthy female dogs. Some blood parameters investigated (neutrophil and lymphocyte count, ALP, PT and APTT) were affected by the use of HBOT.

## 1. Introduction

In veterinary medicine, hyperbaric oxygen therapy (HBOT) involves the administration of 100% oxygen in a pressurized chamber at 1.4–3 absolute atmospheres (ATAs) [1,2,3,4], which can increase up to 28 times the solubility of oxygen in plasma [1,5] to deliver oxygen to different tissues, even without the contribution of hemoglobin [1].

HBOT has been recognized as one of the best adjuvant therapies for the healing of complicated wounds in veterinary [1,3,6] and human medicine [7,8]. Studies have shown that HBOT has applications in preconditioning for surgery in human patients, reducing the rates of postoperative complications and length of stay in the intensive care unit [9]. Additionally, it can provide promising results for the preservation and/or preparation of the surgical bed preoperatively [1,10,11] and assist in the treatment of postoperative wounds [1,6,12,13].

Hemostatic disorders after a surgical event can lead to complications in wound healing due to bleeding, and intraoperative hemorrhagic complications can occur in patients presenting with such conditions [14]. The influence of HBOT on hemostatic parameters has been described in previous studies in rats [15], pigs [16] and humans [17,18]. The influence of HBOT on hemograms has also been observed in previous studies involving human patients [4,19].

Thromboelastography and thromboelastometry are used as global measures of blood coagulation and fibrinolysis in various research and clinical settings, allowing observation of diving-induced hemostatic changes in humans [20]. Nevertheless, hemostasis is a complex process involving dynamic interactions between vessels, platelets and coagulation proteins, with buccal mucosal bleeding time (BMBT) being probably the best measurement of platelet plug formation and function platelet in vivo [21]. BMBT, prothrombin time (PT) and partially activated thromboplastin time (APTT) have not been evaluated in healthy dogs undergoing HBOT, in the same way that there are no studies evaluating the possible effects of HBOT on clinical hemorrhage. Therefore, we sought to verify the effects of a session of HBOT at two ATAs for 45 min on hemograms, serum biochemistry, BMBT, PT and APTT in clinically healthy female dogs undergoing laparoscopic-assisted ovariohysterectomy (OVH). Simultaneously, we sought to verify whether there was a difference in the volume of blood drained during BMBT assessment by measuring the bloodstain area (BA) using a hygroscopic paper. We hypothesized that in healthy female dogs, a session of HBOT can induce significant effects on hemostatic variables immediately after the session, as well as on hemograms and serum biochemistry when followed by laparoscopic-assisted OVH.

## 2. Materials and Methods

Thirty adult mixed-breed female dogs aged 12 to 72 months and weighing 7.1 to 16.7 kg were selected. The inclusion criteria were that the animals were tolerant to handling, healthy at clinical and laboratory examinations and had undergone a focal ultrasound examination of the reproductive tract and out of estrus on vaginal cytology examination. The study protocols, as well as the number of animals used, were approved by the Ethics Committee on the Use of Animals of the Federal University of Santa Maria (protocol no. 7308250522), following all the guidelines of the National Experimentation Control Council Animal. All the owners involved in the project signed an informed consent form.

### 2.1. Experimental Protocol

During the clinical evaluation, patients were randomly allocated to the surgical hyperbaric (HBOT + SURG), hyperbaric (HBOT) and surgical (SURG) groups, with 10 female dogs in each group. The female dogs were admitted for hospitalization and placed in stalls for acclimatization, immediately after collecting a vaginal swab for cytological analysis and defining the phase of the estrous cycle. Commercial dry food and water were provided ad libitum and removed 8 h before the surgical procedure.

Twenty-two hours after admission, patients in the HBOT + SURG and HBOT groups underwent HBOT in a monoplace hyperbaric chamber (Hyperbaric Veterinary Medicine, model H1, Boca Raton, FL, USA) at 2 ATAs (14.7 PSI) for 45 min, with 15 min of pressurization and 15 min of depressurization. During all sessions, an operator previously trained in the Hyperbaric Veterinary Institute (HVI) followed the safety guidelines established by the HVI. The following information was recorded every 5 min by the same operator, who followed each session full-time: behavior of the female dogs; concentrations of O_2_ and CO_2_; and internal pressure, temperature and humidity of the chamber. The flow rate during the session was maintained between 22.6 and 51 L/min (0.8–1.8 cubic feet/min) and the concentration of CO_2_ inside the chamber was monitored full-time through the digital sensors in the chamber itself. In all sessions, the O_2_ concentration after the pressurization period was greater than 90%, and the CO_2_ concentration at the end of the therapy did not exceed 3200 ppm. After HBOT, the dogs in the HBOT group were kept in the preoperative preparation room for 2 h and 30 min before returning to their stalls. Dogs in the HBOT group underwent laparoscopic-assisted OVH only at the end of all study evaluations.

Bitches in the HBOT + SURG and SURG groups were pre-medicated with tramadol hydrochloride (4 mg/kg^−1^, IM) 24 h after admission. A 22-G catheter was positioned in the cephalic vein for drug administration, 15–20 min after premedication. Hair was clipped to the cranial edge of the pubis in the ventral abdominal region, extending laterally to the transverse process of the lumbar vertebrae.

General anesthesia was induced with propofol (4.05 ± 0.22 mg/kg^−1^, IV). The dogs underwent endotracheal intubation maintained with isoflurane in 100% oxygen in a partial gas rebreathing circuit for anesthesia maintenance. After positioning on the surgical table, the dorsal podal artery of one of the pelvic limbs was accessed using a 22-G catheter and connected to a mean arterial pressure measurement system with an analog manometer (Premium, Zhejiang, China). Fentanyl sulfate (5 μg/kg^−1^, IV) was administered via cephalic vein access, and subsequently intraoperative fluid therapy with 0.9% NaCl (10 mL/kg/h) was maintained according to De Oliveira et al. [22], in the same venous access.

### 2.2. Surgical Procedure

Laparoscopic-assisted OVH with two 11 mm ports in the ventral midline was performed as described by Milech et al. [23], with modifications in relation to hemostasis and operative times, by a surgeon proficient in the technique. A pneumoperitoneum of 10 mmHg was maintained at a rate of 1.5 L/min for 30 min in all the female dogs. Manipulation, diathermy hemostasis and cutting maneuvers were performed using 5 mm × 37 cm vascular sealing forceps (LigaSure^®^, Covidien, Miami, FL 33178, USA). Using the LigaSure^®^ forceps, after draining the pneumoperitoneum maintained for 30 min, the right ovary and body of the uterus were exteriorized through the caudal portal wound for hemostasis and section of the mesometrium vessels and the uterine body close to the cervix.

During the surgical procedure, which lasted approximately 57 min, two peritoneal biopsies were collected using 5 mm laparoscopic biopsy forceps as part of a separate study. After drainage of the pneumoperitoneum, abdominorrhaphy was performed using a PDX 2-0 thread in cross-mattress and continuous horizontal mattress patterns for the muscles and subcutaneous tissue, respectively. Wound closure was performed using 4-0 nylon sutures with simple horizontal mattress patterns. Dipyrone (25 mg/kg^−1^, IV) was administered immediately after extubation. One hour after extubation, the female dogs were returned to stalls in the surgery recovery suit and received commercial dry food and water ad libitum, 2 h after extubation.

### 2.3. Blood Samples

With 21-G needles connected to non-heparinized polypropylene syringes, 6 mL of blood was collected from the jugular vein at three different time points: T0, moment of admission; T1, 24 h after T0 (immediately before anesthetic premedication in the SURG group and immediately after HBOT in the HBOT + SURG and HBOT groups); and T2, 48 h after T0 (24 h after HBOT and anesthetic premedication). Mini blood collection tubes were used for EDTA and serum (Labor Import, Guangzhou, China). At T0, 3 mL of blood was collected and immediately divided into three tubes: 0.5 mL in a tube with EDTA (to determine the hemogram); 2 mL in a tube containing 3.2% sodium citrate (to determine the PT, and APTT); and 0.5 mL in a tube without anticoagulant (to determine alanine aminotransferase (ALT), alkaline phosphatase (ALP), creatinine and blood urea nitrogen (BUN)). At T1, 2 mL of blood was collected and immediately stored in a tube containing 3.2% sodium citrate to determine the PT and APTT. At T2, 1 mL of blood was collected and immediately divided into two tubes: 0.5 mL in a tube with EDTA (to determine the hemogram) and 0.5 mL in a tube without anticoagulant (to determine ALT, ALP, creatinine and BUN). In addition to the volume of blood described, another 3 mL of blood was collected at T1, and another 6 mL of blood was collected at two different times for analysis of oxidative status in a separate study.

### 2.4. Processing of Blood Samples

All samples were analyzed within 2 h of collection. The hemogram was determined using an automatic analyzer (Mindray^®^, BC-2800Vet, Animal Medical Technology Co., Ltd., Shenzhen, China). The hematocrit was confirmed in a microhematocrit capillary centrifuged at 8900× *g* for 7 min. The leukogram and platelet count were also confirmed by the differential cell count on a microscope slide and expressed as absolute values per μL. An automatic biochemical analyzer (Mindray^®^, BS-120, Mindray Bio-medical Electronics Co., Ltd., Shenzhen, China) and commercially available veterinary-specific reagents (BioClin^®^, MG, Brazil) were used to determine ALT, ALP, creatinine and BUN in serum samples centrifuged at 900× *g* for 4 min. PT and APTT were determined via the optical colorimetric method with a semi-automatic coagulometer (MaxCoag^®^, MedMAX, São Paulo, Brazil), using plasma, after centrifuging the samples for 15 min at 150× *g*.

### 2.5. Buccal Mucosal Bleeding Time

BMBT was performed as described by Chohan et al. [21], with modifications, immediately after positioning the patient on the surgical table in dorsal recumbency. With the upper lip retracted and a sterile disposable manual lancet (Blood Lancets, Medipoint, Mineola, NY, USA), a vertical incision standardized in length and depth was made at a location without visible evidence of vessels in the mucosa over the second right maxillary premolar. Time counting was started as soon as the mucosa was incised. The drained blood was dried with semicircular hygroscopic paper approximately 1–2 mm from the incision wound, taking care not to touch it. Timing was maintained until there was no further bleeding from the incision and the BMBT was recorded in seconds.

Images of the hygroscopic paper were obtained using a scanner (HP, Scanjet G4050, Hewlett-Packard, Livermore, CA, USA) with a resolution of 300 DPI (2574 × 3696 pixels) in JPEG, RGB, eight-bit format, using the HP Scanjet Scanner software for Microsoft Windows^®^7. To calculate the BA on the hygroscopic paper, we used ImageJ2 version 2.14.0, as previously described [24], with modifications. All images were analyzed using the algorithm shown in Figure 1. BA was recorded in mm^2^.

### 2.6. Postoperative Management

Approximately 2.5 h after T1, all the female dogs were kept in stalls with a surgical recovery suit, commercial dry food and water ad libitum for 16 h. Outdoor walks were offered every 6 h. Patients in the HBOT + SURG and SURG groups were treated with dipyrone (25 mg/kg^−1^, SC) and tramadol hydrochloride (4 mg/kg^−1^, SC) every 8 h. The HBOT group received 0.9% NaCl in a volume corresponding to the medications used in the surgical groups via the subcutaneous route.

At hospital discharge, dogs in the HBOT + SURG and SURG groups were prescribed a surgery recovery suit until the skin suture was removed within 9 days, in addition to dipyrone (25 mg/kg^−1^, PO, every 8 h) and tramadol hydrochloride (4 mg/kg^−1^, PO, every 8 h) for 4 days. At hospital discharge of the dogs in the HBOT group, the same drugs and respective doses were prescribed for 5 days, in addition to the use of a surgery recovery suit until the skin suture was removed within 10 days.

### 2.7. Statistical Analysis

The sample size was determined via the one-way fixed-effects ANOVA test in a statistical program (G Power^®^, version 3.1.9.6, Heinrich-Heine-Universität Düsseldorf, Düsseldorf, Germany) using a test power of 0.8 with a significance level of 0.05 and an effect size of 0.6. Microsoft Excel 2018 (Microsoft^®^, version 16.16.14, Redmond, Washington, DC, USA) was used for descriptive analysis and table creation. Data between groups and time points were compared using the statistical program ActionStat version 3.6.331.450 (Estatcamp, São Carlos, Brazil). The Shapiro–Wilk test was used to evaluate the normal distribution of data, expressed as arithmetic mean ± standard deviation, except for non-parametric data, which were expressed as median (range). Student’s *t*-test was used to compare parametric data. The Wilcoxon independent samples test was used to compare non-parametric data. Statistical significance was set at *p* < 0.05.

## 3. Results

### 3.1. Clinical Assessment and Screening

Of the 50 female dogs accepted for clinical screening, 12 were rejected because of changes in the reproductive tract, four because they showed changes in the vaginal cytology examination, and four because of changes in the hemogram. Thirty female dogs were included in the study with an average weight of 11.1 ± 2.7 kg. The average total surgical time was 57.4 ± 2.04 min, with no difference between the HBOT + SURG and SURG groups (*p* = 0.6580). There was no difference in mean arterial pressure values (*p* = 0.1216) between the HBOT + SURG (69.7 ± 5.54 mmHg) and SURG (76.9 ± 12.7 mmHg) groups. Minor adverse effects such as shivering, panting, vocalization and ear flicking were observed in approximately half of the dogs undergoing HBOT. No major adverse effects occurred during the HBOT sessions, nor were there any transoperative or perioperative surgical complications.

### 3.2. Hemogram and Serum Biochemical Profile

When comparing groups, at T0, there was no statistical difference in the values of the variables evaluated in the hemogram (*p* ≥ 0.1275) (Table 1). At T2, a higher neutrophil count was observed in the HBOT + SURG and SURG groups than in the HBOT group (*p* ≤ 0.0324) (Table 1).

When comparing times, within each group, for hemogram values, in the groups undergoing surgery (HBOT + SURG and SURG), there was an increase in the neutrophil count (*p* ≤ 0.0039). In the SURG group, there was an increase in total leukocyte count (*p* = 0.0238). In the HBOT + SURG group, there was a reduction in the lymphocyte count (*p* = 0.0115) (Table 1).

When comparing the groups, at T0, there was no statistical difference for the values of the variables evaluated in the biochemical profile (*p* ≥ 0.1128), except for ALP between the HBOT + SURG and HBOT groups (*p* = 0.0256) (Table 2). Lower ALP values in the HBOT group compared to the HBOT + SURG and SURG groups were observed at T2 (*p* ≤ 0.0493) (Table 2). When comparing times, within each group, for biochemical profile values, a higher ALP activity was observed in the HBOT + SURG group at T2 compared to T0 (*p* = 0.0233) (Table 2).

### 3.3. Prothrombin Time and Activated Partial Thromboplastin Time

When comparing groups, the HBOT + SURG group presented higher APTT values compared to the HBOT and SURG groups at T0 (*p* ≤ 0.0355). There was no statistical difference between the groups for PT values at T0 (*p* ≥ 0.1655) (Table 3). When comparing groups, at T1, there was no statistical difference for PT (*p* ≥ 0.4359) and APTT (*p* ≥ 0.4272) values (Table 3).

When comparing time within each group, a reduction in PT (*p* = 0.0089) and APTT (*p* = 0.0412) values was observed in the HBOT + SURG group. When comparing time, in the other groups, no significant variation was observed in PT (*p* ≥ 0.123) and APTT (*p* ≥ 0.6842) values (Table 3).

### 3.4. Buccal Mucosal Bleeding Time and Bloodstain Area (BA)

There was no significant difference between the BMBT values of the HBOT + SURG (55.5 s, range 40–73 s) and SURG (57 s, range 39–165 s) groups (*p* = 0.6769). There was no significant difference between the BA values of the HBOT + SURG (35.5 ± 48 mm^2^) and SURG (59.3 ± 79.4 mm^2^) groups (*p* = 0.1697).

## 4. Discussion

### 4.1. Invasive Blood Pressure

The literature provides evidence that HBOT attenuates the decrease in mean arterial pressure in rats with multiple organ failure [15] and induces an increase in parasympathetic tone with increased peripheral vascular resistance, redistribution of peripheral to central blood flow and bradycardia in professional divers [25]. During peripheral vasoconstriction, there is an increase in systemic blood pressure and, consequently, a reduction in heart rate [1,13].

According to Schipke et al. [26], bradycardia in humans undergoing HBOT occurs via the baroreceptor reflex in response to increased blood pressure and is not directly owing to an increase in parasympathetic tone. The same authors mentioned that endothelial injury owing to oxidative stress reduces the concentration of endothelial vasodilators, leading to vasoconstriction, which corroborates previous studies stating that such vasoconstriction is induced by the inhibition of nitric oxide [6,27] and prostaglandin synthesis [6,28].

In our study, we did not observe any difference in mean arterial pressure values (*p* = 0.1216) between the HBOT + SURG (69.7 ± 5.54 mmHg) and SURG (76.9 ± 12.7 mmHg) groups; however, our study evaluated this parameter during the surgery of healthy dogs, that is, after the hyperbaric session under the effects of anesthetic drugs.

### 4.2. Erythrocyte and Platelet Count

We did not observe significant changes in the erythrogram and platelet count values in our study. In veterinary literature, studies evaluating the influence of HBOT on erythrocyte count, hemoglobin and hematocrit are scarce. In a feline patient who received 37 sessions of HBOT as adjuvant therapy for aortic thromboembolism, Reinstein et al. [10] observed an increase in platelet count and a reduction in hematocrit after 25 days of daily therapy, followed by an increase in hematocrit and leukopenia until the last session on the 51st day.

Studies that have evaluated erythrograms in humans have yielded varying results. Sinan et al. [29] observed a reduction in hematocrit and erythrocyte counts in patients who received HBOT after 20 sessions, without any change after the first session. Bosco et al. [19] observed that athletes’ undergoing 20 HBOT sessions on alternate days contributed to elevated hemoglobin levels. However, Gunes and Aktas [4] conducted a study involving 140 human patients indicated for HBOT undergoing up to 60 sessions, where a temporary reduction in platelet count was observed without any effect of HBOT on the values of red blood cells, hemoglobin and hematocrit, corroborating a previous study that described similar results in healthy humans [30].

It is worth considering that the evaluation of the effects of HBOT on hematological parameters in retrospective observational studies presents a bias towards involving patients with inflammatory comorbidities, which implies changes in hematopoiesis, making HBOT one of the least likely factors for changes in the hemogram [4]. However, the literature cites that HBOT reduces the levels of cytokines interleukin-1 and tumor necrosis factor-alpha, which suppress the growth of erythroid cells in cases of chronic inflammation [4].

### 4.3. White Blood Cell Count

Of the leukogram values, only the neutrophil count showed a clinically significant variation at T2 in the HBOT + SURG and SURG groups, which was different from the total leukocyte and lymphocyte counts, which did not show a clinically significant variation [31]. In our study, we used a pneumoperitoneum of 10 mmHg at 1.5 L/min for 30 min, which induces ischemia and reperfusion in veterinary and human patients, as described previously [23,32,33]. Increased intra-abdominal pressure leads to decreased blood flow through the splanchnic vessels with relative hypoperfusion of the celiac, superior mesenteric and renal arteries, despite normal blood pressure [34]. In general, reperfusion occurs after normalization of the intra-abdominal pressure. 

Therefore, laparoscopy can be considered a model of ischemia-reperfusion [32]. Thus, the laparoscopic-surgical procedure, although it provides reduced tissue damage, may explain the neutrophilia observed in the surgical groups in relation to the HBOT group (*p* ≤ 0.0324), as well as the lack of statistical difference between the HBOT + SURG and SURG groups at T2 for the neutrophil count (*p* = 0.9996), since an increase in neutrophil count is expected in canine patients undergoing elective OVH [32]. According to Ortega et al. [35], among the most important immunomodulatory effects of HBOT, the activation of neutrophils with migration to hyperoxic regions and the reduction of lymphocytes stand out. Reducing the adhesion and sequestration of neutrophils by tissues is an immunomodulatory effect of HBOT observed in several laboratory models that attenuate ROS formation, especially during the reperfusion phase [13,36].

Within physiological values [31], an increase in total leukocyte count observed in the SURG group (*p* = 0.0238) was expected after conventional or laparoscopic-surgical procedures [32]. In our study, we did not observe a significant change in the lymphocyte count 24 h after laparoscopic OVH in healthy dogs.

The reduction in lymphocyte count observed in the HBOT + SURG group at T2 compared with that at T0 (*p* = 0.0115), even within physiological values [31], corroborates what has been described in the literature in humans and rats [35,37]. In a study on Wistar rats with acute pancreatitis, Bai et al. [37] reported that treatment with HBOT can lead to a significant reduction in the serum levels of pro-inflammatory cytokines (IL-2 and INF-c) and in the proliferation of Th1 cells (secreting pro-inflammatory cytokines), leading to a reduced lymphocyte count. In our study, the increase in total leukocyte count observed in the SURG group between T0 and T2 was not observed in the HBOT + SURG group (*p* = 0.1925), which may be related to the reduction in lymphocyte count observed in the HBOT + SURG group between T0 and T2.

### 4.4. Serum Biochemistry

Of the four biochemical markers evaluated in our study, we observed significant variations only in ALP, which increased in the HBOT + SURG group at T2 within the physiological range according to the literature [38]. The smaller ALP value at T0 in the HBOT group compared to the HBOT + SURG group becomes a bias for the comparison of the values of this enzyme between these groups. Additionally, studies have reported variations in ALP activity caused by intestinal injury in rats [39] and greater osteoblast activity in humans undergoing HBOT [7]. However, no studies have evaluated the influence of HBOT on serum biochemistry in dogs.

In a study involving Wistar rats, Oter et al. [39] observed a reduction in ALT and ALP activities in animals subjected to sepsis and treated with antibiotic therapy plus HBOT, but without significant changes in these markers among healthy animals subjected only to HBOT. Another study involving forty-four Sprague–Dawley rats with septic peritonitis used ALT as a sensitive indicator of hepatotoxicity and observed a reduction in serum ALT activity in the group undergoing HBOT compared to the control group [40].

Regarding the effects of HBOT on renal serum biochemical markers, the literature observes, in models of renal ischemia injury in rats, a reduction in serum creatinine and IL-6 values [41], as well as in BUN and proteinuria values [42], indicating the beneficial role of HBOT in preventing ischemia and reperfusion injury in this species. However, the literature describes that such benefits are not observed when HBOT is used in combination with certain nephrotoxic medications, such as cisplatin and gentamicin [43].

### 4.5. Prothrombin Time and Activated Partial Thromboplastin Time

The experience of our team in a previous study led us to suspect a higher incidence of bleeding in skin accesses of healthy felines undergoing HBOT [44]. However, in the veterinary literature, studies investigating the effects of HBOT on hemostatic parameters are scarce and restricted to rats and pigs [15,16]. Furthermore, some studies in humans have yielded controversial results.

Peng et al. [20] observed a reduction in the time required for clot formation in trained naval divers 75 min and 5 h after they were subjected to decompressive stress in a hyperbaric chamber. These results corroborate those of a study by Monaca et al. [18], who observed induction of the procoagulatory axis in humans with underlying disease who underwent a session of HBOT at 2.4 ATAs for 90 min. The literature also observed a reduction in APTT in humans [18], which corroborates our findings, as we observed a reduction in both PT and APTT in the HBOT + SURG group at T1 compared to T0. The higher APTT value at T0 for the HBOT + SURG group in relation to the other groups became a bias for comparing the values of this variable; however, there was still no difference between the groups.

Different results have been described by Imperatore et al. [15], who observed that one HBOT session at 2 ATAs for 60 min in rats induced multiple organ failure syndrome; attenuated the activation of the coagulation system (PT, APTT, and fibrinogen); and attenuated the activation of fibrinolysis, thrombocytopenia and platelet hyperaggregation. Miike et al. [17] described a reduction in adherence and clot formation capacity in blood samples from humans with necrotizing fasciitis, skin ulcer and soft tissue ulcers subjected to three 60 min HBOT sessions at 2 ATMs.

Although crystalloids can dilute coagulation factors, hypercoagulability can also be observed [21] because of the probable dilution of anticoagulant factors [45] as well as an increase in BMBT and a reduction in platelet count without reducing platelet aggregation [21]. However, the literature refers to blood dilutions above 20% of 0.9% NaCl in the blood, that is, in 1:4 mixtures [45], which is an extremely high volume compared to that used in the surgical groups in our study, with fluid therapy introduced at 10 mL/kg/h [22] for an average total surgical time of 57.4 ± 2.04 min.

### 4.6. Buccal Mucosal Bleeding Time and Bloodstain Area

Platelet count, PT, APTT and BMBT are described as some of the main methods for evaluating coagulation [14,21]. In our study, as none of the BMBT and BA assessment results differed between the HBOT + SURG and SURG groups, our results do not support the hypothesis that HBOT affects bleeding in healthy canines undergoing HBOT.

### 4.7. Limitations of the Study

In healthy female dogs subjected to OVH, Dalmolin et al. [32] observed, for some of the hemogram parameters, more significant variations at 6 and 12 h after the surgical event than at 24 h postoperatively. Therefore, the hypothesis that a period of greater variation in neutrophil counts may have been incorporated into our study should be considered. Studies in humans showed significant changes in hematocrit and erythrocyte counts only after 20 sessions of HBOT [19,29], which suggests the possibility that we did not include enough time in our study to observe such changes. 

We observed a reduction in both PT and APTT in the HBOT + SURG group when comparing T1 and T0. However, the same did not occur in the HBOT group, which could explain the discrepancy between these results. Because there were differences in the baseline values of ALP and APTT between some of the groups studied, a study with a more homogeneous population is necessary to evaluate these variables.

Despite this, our objective was precisely to understand and define the immediate changes in the studied parameters associated with or not associated with a single preoperative HBOT session. Therefore, we consider our study to be a small contribution to the search for definitions of the advantages and disadvantages of using HBOT shortly before surgical procedures in dogs. Therefore, we opted for elective surgery in healthy animals.

Despite the period of acclimatization to the management and hospitalization stalls, the different physiological responses to stress induced in patients undergoing HBOT and/or surgical procedures may have influenced the laboratory parameters investigated. Future prospective studies should consider the use of a formal anxiety scoring system.

## 5. Conclusions

We conclude that a session of HBOT at two ATAs for 45 min does not induce significant effects on mean arterial pressure, BMBT and BA in healthy female dogs undergoing elective surgery proposed in this model, as well as not inducing significant effects on the erythrogram, platelet count, WBC, neutrophils, monocytes, eosinophils, ALP, ALT, BUN and creatinine. A reduction in PT and APTT was observed just immediately after the HBOT session, as well as a reduction in the lymphocyte count after HBOT when followed by laparoscopy-assisted OVH. Future studies involving a more homogeneous population and a greater and more extensive collection frequency involving the postoperative period will allow for more conclusive results regarding the hemostatic effects of HBOT in canine patients undergoing surgery.

## Figures and Tables

**Figure 1 animals-14-01785-f001:**
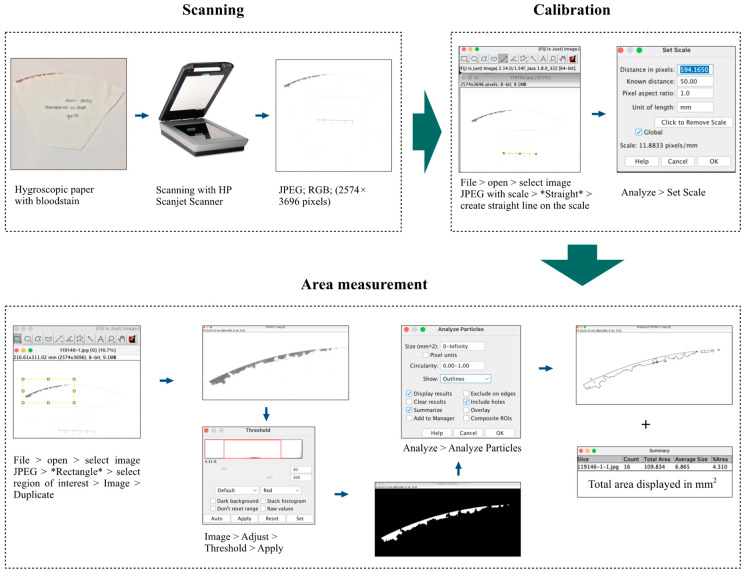
Flow diagram showing the image analysis methodology using software to measure bloodstain on hygroscopic paper. The methodology starts from the digitization of hygroscopic papers using the HP Scanjet G4050 equipment, along with a known size scale; image import into the ImageJ2 program; determination of image scale size; selection of the region of interest; establishing specific color threshold settings for the bloodstain; finalizing recording the measurement using the “Analyze Particles” function.

**Table 1 animals-14-01785-t001:** Means ± standard deviation (SD) hemogram values in total blood of the canine study population, according to the belonging group and collection time.

Variable	HBOT + SURG		HBOT		SURG	
	T0	T2	T0	T2	T0	T2
RBC	7.06 ± 0.79	6.88 ± 0.83	7.37 ± 0.55	6.96 ± 0.53	6.71 ± 0.81	6.34 ± 0.88
Hgb	15.74 ± 1.72	15.42 ± 1.99	16.97 ± 2.03	16.07 ± 1.53	15.27 ± 1.89	14.82 ± 2.22
HCT	47.28 ± 6.17	45.94 ± 5.91	50.73 ± 5.22	47.82 ± 4.81	46.8 ± 5.29	44.61 ± 6.29
PLT	250.5 ± 62.96	265.6 ± 112.42	255 ± 101.32	244.6 ± 66.15	229.5 ± 47.37	252.1 ± 72.84
WBC	11.33 ± 4.85	14.36 ± 4.17	11.36 ± 2.92	12.13 ± 2.68	11.4 ± 2.99 ^1^	15.12 ± 3.71 ^2^
Neu	6.99 ± 1.66 ^1^	11.47 ± 4.38 ^a,2^	7.12 ± 2.26	7.75 ± 1.9 ^b^	7.04 ± 1.8 ^1^	11.03 ± 2.46 ^a,2^
Lym	3.28 ± 1.15 ^1^	2.19 ± 0.43 ^2^	2.62 ± 1.01	2.69 ± 0.68	2.42 ± 0.82	2.46 ± 1.11
Mon	0.51 ± 0.26	0.86 ± 0.45	0.38 ± 0.27	0.63 ± 0.44	0.55 ± 0.33	0.64 ± 0.39
Eos	1.24 ± 1.09	1.10 ± 0.94	1.22 ± 0.89	1.04 ± 0.82	1.39 ± 1.44	0.99 ± 1.05

^a,b^ Different letters indicate significant differences between treatments at the same evaluation time. ^1,2^ Different numbers indicate significant differences between evaluation times. RBC: red blood cell (cell × 10^6^/μL); Hgb: hemoglobin (g/dL); HCT: hematocrit (%); PLT: platelet (cell × 10^3^/μL); WBC: white blood cell (cell × 10^3^/μL); Neu: neutrophil (cell x 10^3^/μL); Lym: lymphocyte (cell × 10^3^/μL); Mon: monocyte (cell × 10^3^/μL); Eos: eosinophil (cell × 10^3^/μL). HBOT + SURG, hyperbaric surgery group; HBOT, hyperbaric group; SURG, surgery group.

**Table 2 animals-14-01785-t002:** Medians (range) of liver and kidney serum biochemical values of the canine population studied, according to group of belonging and collection time.

Variable	HBOT + SURG		HBOT		SURG	
	T0	T2	T0	T2	T0	T2
ALT	36 (24–124)	36 (22–79)	39 (23–94)	38.5 (22–111)	29 (17–55)	36 (25–52)
ALP	53 (40–108) ^a,1^	72 (50–109) ^a,2^	38.5 (18–125) ^b^	40.5 (22–116) ^b^	48 (21–80)	53.5 (34–155) ^a^
BUN	28.5 (10–46)	30 (12–42)	27 (15–54)	30.5 (23–42)	23.5 (13–68)	26.5 (15–42)
Crea	0.9 (0.8–1.2)	0.9 (0.7–1.2)	0.8 (0.6–1)	0.8 (0.7–1.1)	0.9 (0.3–1.2)	1 (0.7–1)

^a,b^ Different letters indicate significant differences between treatments at the same evaluation time. ^1,2^ Different numbers indicate significant differences between evaluation times. ALT, alanine aminotransferase (UI/L); ALP, alkaline phosphatase (U/L); BUN, blood urea nitrogen (mg/dL); Crea, creatinine (mg/dL). HBOT + SURG, hyperbaric surgery group; HBOT, hyperbaric group; SURG, surgery group.

**Table 3 animals-14-01785-t003:** Medians (range) for hemostatic variables of the canine population studied, according to group of belonging and collection time.

Variable	HBOT + SURG		HBOT		SURG	
	T0	T1	T0	T1	T0	T1
PT	13.5 ^1^ (9.7–29.7)	11.1 ^2^ (9.8–15.4)	12.6 (8.4–26.1)	11.3 (10.3–13)	11.4 (9.6–22.5)	11.9 (7.8–19)
APTT	15.5 ^a,1^ (13.5–21.3)	13.8 ^2^ (11.1–17.5)	13.5 ^b^ (11–18.6)	13.1 (12.4–16.3)	14.1 ^b^ (11.6–15.6)	14.1 (11.6–20.5)

^a,b^ Different letters indicate significant differences between treatments at the same evaluation time. ^1,2^ Different numbers indicate significant differences between evaluation times. PT: prothrombin time (s); APTT: activated partial thromboplastin time (s). HBOT + SURG, hyperbaric surgery group; HBOT, hyperbaric group; SURG, surgery group.

## Data Availability

Publicly available datasets were analyzed in this study. These data were obtained by contacting the corresponding author via email.

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
