# Peer review of "Effects of Hyperbaric Oxygen Therapy on Hemogram, Serum Biochemistry and Coagulation Parameters of Dogs Undergoing Elective Laparoscopic-Assisted Ovariohysterectomy"

_animals, 2024, doi:10.3390/ani14121785_

Round 1

Reviewer 1 Report

Comments and Suggestions for Authors

Dear Authors

The authors evaluate Effects of hyperbaric oxygen therapy on blood count, serum biochemistry and intraoperative bleeding parameters of female dogs undergoing elective video-assisted ovariohysterectomy.

I believe that manuscripts are not published in VS.

The result does not suggest new scientific information.

We divided 3 groups (HBOT+SURG group (exposure to 2 ATA for 45 min followed by video-assisted OVH), HBOT group (exposure to 2 ATA for 45 min) and SURG group (video-assisted OVH). We need to evaluate other groups for the effect of hyperbaric oxygen therapy (HBOT) in clinical fields.

Sincerely yours.

Author Response

Dear Reviewer, on behalf of the authors, we thank you for your valuable reviewer considerations that make the article more objective and clear. We hope to have answered the main questions proposed. In the attached document, below each comment made by the reviewer we have the responses in blue.

Reviewer 2 Report

Comments and Suggestions for Authors

I appreciate the opportunity to evaluate this manuscript. The paper of Antunes et al. evaluates the effects of hyperbaric oxygen on the hemogram, serum biochemistry and coagulation parameters in bitches undergoing elective video-assisted OVH. The paper presents interesting results, however, in my point of view, there are some major concerns:

Title:

As coagulation parameters are not evaluated intraoperatively I suggest to change the title. For example, "Effects of hyperbaric oxygen therapy on the hemogram, serum biochemistry and coagulation parameters of bitches undergoing elective video-assisted OVH ".

Simple summary

Should be reviewed and improve for english.

The acronym "ATA" should be previously defined

Abstract

The acronym "ATA" should be previously defined

Key words: Please remove "monoplace", this information is not given in the manuscript.

Introduction

In this section and throughout the manuscript "blood count" would better be replaced by "Hemogram".

Lines 64 to 66. Regarding the sentence "... with buccal mucosal bleeding time... platelet plug formation an function platelet in vivo" It's best not to forget the Thromboelastography (TEG) and thromboelastometry (ROTEM) mainly used in surgery and anesthesiology. This sentence is not only somewhat inaccurate, but the technique described is more invasive than others. Rewrite the introduction based in this guideline.

At no point in the introduction do the authors refer to the bloodstain area in a hygroscopic paper (BA) and its usefulness in accordance with the proposed objectives. The purpose of its use should be explained, or alternatively as this methodology did not produce statistically significant and clinically relevant results, consider to remove this information from the manuscript.

M&M

Line 95/96 - Please refer O2 and CO2 chamber concentrations.

How did you control the CO2 accumulation in the chamber?

Was there any camber operator certification?

What safety measures were taken?

Patients were sedated prior to entering the chamber?

Was there any type of monitoring of the animal during treatment?

Any side effects have been reported?

Line 98 - The acronym "CFM" should be previously defined

Line 140 - The acronym "MPA" should be previously defined

Lines 146/147 - Authors should make clear the chosen biochemistry panel

Lines 156/157 -Please change "blood count, leukogram and platelet count" to "Hemogram"

Lines 159, 164 and 167, - Centrifugation force should better be set in "G" force

Results

Line 232 and 247 - Please change "segmented neutrophil" to "neutrophil"

Discussion

Line 312 - Please change "operation" to "surgery"

Line 323 - Please change "herithrogram" to "erythrogram"

Line 343 - The described experiment by Cronin and co-workers seems to be incomplete.

Lines 351 /353 - It seems out of context to start the discussion regarding the white blood cell count with this phrase regarding to laparoscopy Line 358 - Please change "blood count" to "leukogram" Line 439 - consider to remove discussion related to bloodstain area.

Comments on the Quality of English Language

Authors should review the English of their manuscript especially with regard to the simple summary

Author Response

Dear Reviewer, on behalf of the authors, we thank you for your precious reviewer's considerations which make the article more objective and clear. We hope we have met the necessary changes and bring all explanations of the changes attached. In the attached document, below each comments made by the reviewer we bring in blue the alterations/answers. We highlight changes to the manuscript within the copy-edited version in the platform using red text.

Reviewer 3 Report

Comments and Suggestions for Authors

REVIEW REPORT

Paper “Effects of hyperbaric oxygen therapy on blood count, serum biochemistry and intraoperative bleeding parameters of female dogs undergoing elective video-assisted ovariohysterectomy”

Authors: Antunes B.N. et al.

The authors described the effects of a single session of hyperbaric oxygen therapy on hematological parameters in healthy female dogs prior to video-assisted ovariohysterectomy.

The manuscript is clear and well written. The topic is interesting. HBOT is widely used in human medicine, but it is still little used in veterinary medical practice.

It is known that HBOT is useful as adjunctive treatment of complicate wounds and injuries, but it is interesting to better investigate his effects on healthy animals undergoing surgical treatments. It is also interesting to find biomarkers that could quantify the physiological effects of hyperbaric oxygen therapy on human and animal patients.

I suggest the following modifications:

1)      Simple summary and abstract are very similar. The authors should rewrite the simple summary and describe briefly properties (immunomodulatory, antimicrobial, angiogenetic, etc… properties) and clinical applications of HBOT and the aim of their study. It is not necessary to cite methods and results of the study, that are described in the abstract.

2)      Materials and methods, lines 95-97. Authors stated that the behaviour of the bitches were recorded during HBOT session. In lines 227-228 they stated that no complications or surgical complications occurred during the study. Did the authors observed side effects on dogs during or after HBOT session?  According to literature, oxygen could be toxic for the CNS. Authors should add a sentence in paragraph 3.1 to explain if any side effects of HBOT were recorded.

3)      Line 52: replace “HOBT” with “HBOT”.

4)      Line 140: what is the expanded version of the acronym “MPA”? It was not mentioned in the text.

5)      Line 164: replace “FA” with “ALP”.

6)      Line 165: replace “TP” with “PT”.

7)      Were dipyrone and tramadol hydrochloride prescribed for dogs in the surgical and HBOT+SURG groups? If yes, replace “HBOT” (line 204) with “HBOT+SURG”.

8)      Line 240: replace “RCB” with “RBC”.

Author Response

Dear Reviewer, on behalf of the authors, we thank you for your precious reviewer's considerations which make the article more objective and clear. We hope to have answered the main questions proposed. In the attached document, below each comments made by the reviewer we bring in blue the answers. We highlight changes to the manuscript within the copy-edited version in the platform using red text.

Reviewer 4 Report

Comments and Suggestions for Authors

Dear Authors,

I really appreciate your tremendous effort you did in this study to submit the manuscript. By my standpoint, you worked hard but the manuscript to be accepted for publication I consider reasonable to be re-write properly. Please bear in mind just a few remarks (not an exhaustive list):

-        Major deficiencies: 1. the study has no hypothesis

                                          2. there are a lot of surgical terms which are not appropriate (ex. dermorrhaphy, surgical clothing?, trichotomy )

                                          3. inappropriate and poor scientific language                                      

-        Please re-write the Simple Summary (rows 14-26), it content is inserted identical more than 90% in the content of Abstract (rows 27-42)

-        Row 52, replace preconditioning ” with „ preparing ” for surgery

-        Row 74, replace ”„ of no defined breed ” with „ crossbreed or mixbreed

-        Row 87 replace „ received ” with „ admitted

-        Rows 134,195(and others many rows)  replace the „ surgical clothing ” with proper terms

Comments on the Quality of English Language

-

Author Response

(The authors gave the same response as above.)

Round 2

Reviewer 1 Report

Comments and Suggestions for Authors

Dear. the Authors

The authors evaluate “Effects of hyperbaric oxygen therapy on blood count, serum biochemistry and intraoperative bleeding parameters of female dogs undergoing elective video-assisted ovariohysterectomy” .

This paper is well organized, and I evaluated the author’s efforts highly.

I think that this manuscript can be accept.

Sincerely yours.

Author Response

Dear Reviewer,

As promised, the English writing has been revised. We are very grateful for the reviewer's valuable considerations and are happy to know that we responded appropriately to improve our work.

We thank you for your attention and time dedicated to our work.

Best regards,

Reviewer 2 Report

Comments and Suggestions for Authors

I am grateful to the authors for accepting the vast majority of the suggestions made. It is my opinion that the manuscript is now quite improved. However, there are still minor corrections to be made.

When correcting the rotations for "g" you must eliminate the information in RPM as it becomes confusing.

Line 102 - Explain better what it is and where is located the Hyperbaric Veterinary Institute.

Line 109 - According to your explanations, please add "The concentration of CO2 inside the chamber is monitored full time through the digital sensors in the chamber itself."

Line 228 - According to your explanations, please add "In approximately half of patients undergoing HBOT, minor adverse effects were observed, such as: shivering, panting, vocalization and ear flick."

Lines 156/7 - Please explain de choice of biochemistry parameters. Why these? what are you seeking?

Line 340 - Please change " observed in athletes that twenty HBOT sessions on alternate days"

to "observed in athletes that underwent twenty HBOT sessions on alternate days".

Line 358 - The described experiment by Cronin and co-workers is still incomplete. What happened to platelets?

Comments on the Quality of English Language

English could be improved.

Author Response

Dear Reviewer,

As promised, the English writing has been revised. We are very grateful for the reviewer's valuable considerations and are happy to know that we responded appropriately to improve our work.

We thank you for your attention and time dedicated to our work.   Please see the attachment   Best regards,

Reviewer 4 Report

Comments and Suggestions for Authors

Dear Authors,

I really appreciate your tremendous effort you did to revise the manuscript. By my standpoint, you worked hard and  the manuscript is ready to be accepted for publication.

Author Response

Dear Reviewer,

As promised, the English writing has been revised. We feel very happy to receive this from you. We are grateful for your precious and dedicated contribution to improving our work.

Best regards,
